# Köbner and Pastia Signs in Acute Hemorrhagic Edema of Young Children: Systematic Literature Review

**DOI:** 10.3390/children9020265

**Published:** 2022-02-15

**Authors:** Gabriel Bronz, Danilo Consolascio, Mario G. Bianchetti, Pietro O. Rinoldi, Céline Betti, Sebastiano A. G. Lava, Gregorio P. Milani

**Affiliations:** 1Pediatric Institute of Southern Switzerland, Ospedale San Giovanni, Ente Ospedaliero Cantonale, 6500 Bellinzona, Switzerland; mario.bianchetti@usi.ch (M.G.B.); po.rinoldi@gmail.com (P.O.R.); celine.betti@eoc.ch (C.B.); 2Family Medicine Institute, Faculty of Biomedical Sciences, Università della Svizzera Italiana, 6900 Lugano, Switzerland; danilo.consolascio@students.unibe.ch; 3Pediatric Cardiology Unit, Department of Pediatrics, Centre Hospitalier Universitaire Vaudois, University of Lausanne, 1011 Lausanne, Switzerland; webmaster@sebastianolava.ch; 4Pediatric Unit, Fondazione IRCCS Ca’ Granda Ospedale Maggiore Policlinico, 20122 Milan, Italy; milani.gregoriop@gmail.com; 5Department of Clinical Sciences and Community Health, Università degli Studi di Milano, 20122 Milan, Italy

**Keywords:** acute hemorrhagic edema, cockade purpura, Finkelstein–Seidlmayer syndrome, infantile Henoch–Schönlein syndrome, Köbner sign, leukocytoclastic vasculitis, Pastia sign

## Abstract

Acute hemorrhagic edema of young children, a benign skin-limited vasculitis, predominantly affects children 2 years of age or younger. The prevalence and clinical features of the Köbner and Pastia signs have never been systematically investigated in this vasculitis. To address this issue, we analyzed the data contained in the Acute Hemorrhagic Edema Bibliographic Database, which incorporates all reports on hemorrhagic edema published after 1969. A total of 339 cases (236 males and 103 females; 11 (8–18) months of age; median and interquartile range) were documented with at least 1 photograph and therefore included in this analysis. The Köbner sign occurred in 24 cases (14 males and 10 females; 11 (7–17) months of age), the Pastia sign in 51 cases (39 males and 12 females; 11 (8–15) months of age), and both Köbner and Pastia signs in 8 cases (7 males and 1 female; 11 (7–17) months of age). The lower legs, thighs, waistline, and groin were the regions that were most commonly affected with the Köbner sign, while the ankle, feet, cubital fossa, and elbow were most affected with the Pastia sign. The Köbner and Pastia signs are clinically relevant; they occur in about every fourth child affected with hemorrhagic edema and do not influence the disease progression.

## 1. Introduction

Acute hemorrhagic edema of young children (AHE), often referred to as acute cockade purpura and edema, Finkelstein–Seidlmayer syndrome or infantile Henoch–Schönlein syndrome, is a skin-limited small-vessel leukocytoclastic vasculitis [1,2,3,4] that typically occurs after a simple, mostly viral, febrile illness (more rarely after a vaccine). It predominantly affects children 4 weeks to 2 years of age, spontaneously remits within 3 weeks and does not recur [1,2,3,4]. Recent data point out that a mild and self-remitting abdominal, articular or renal involvement occurs in about 15% of cases [3]. This condition was initially reported in 1913 in the USA by I. Snow, in 1925 by M. Landsberger in Germany, and later in 1936 in Argentina by M. Del Carril. The most comprehensive descriptions, however, were made in Germany before the Second World War: in 1929 by H. Finkelstein (1865–1942) and in 1939 by H. Seidlmayer (1910–1965). A noteworthy description was also made in 1942 by M. Lelong, a French pediatrician [2].

Non-blanching linear skin lesions occasionally occur in apparently healthy skin following mechanical friction. This tendency, habitually termed the Köbner sign, is frequent and well-known in vitiligo, psoriasis, lichen planus, and many bullous dermatoses [5]. On the other hand, pink or red lines produced by confluent petechiae in the creases of the skin, usually termed the Pastia sign, are frequently observed in the context of scarlet fever or, less frequently, eruptions caused by Mycoplasma pneumoniae and pediatric inflammatory multisystem syndrome [6,7]. Available reviews do not or only marginally mention the existence of the Köbner sign in the setting of skin-limited or -predominant vasculitis, except for a narrative review suggesting a prevalence of about 10% [8]. Furthermore, very recent data insinuate that the Köbner sign occurs in about one percent of patients with AHE [3]. Finally, the Pastia sign was never reported in this vasculitis apart from an infant with AHE recently published as a dermatology quiz [9].

The data contained in the **A**cute **H**emorrhagic **E**dema **BI**bliographic **D**atabase **AHEBID** [3,4,10] were employed to investigate the prevalence, characteristics, and clinical relevance of the Köbner and Pastia signs in AHE.

## 2. Materials and Methods

### 2.1. Search Strategy

AHEBID was created in 2019 and incorporates the original articles on AHE published after the original report in 1913 [3,4,10]. The database is obtainable on request (email: finkelstein-seidlmayer@usi.ch). For this purpose, the bibliography search engines Excerpta Medica, the National Library of Medicine databases and Google scholar are screened every second month using the 2020 PRISMA guidelines and checklist [10] (Figure 1) for “acute hemorrhagic edema”, “cockade purpura and edema”, and “infantile Henoch–Schönlein purpura” without any language restriction. AHEBID also includes the original literature on hemorrhagic edema collected by some of us in the early eighties of the last century.

As of 1 July 2021, the database included 317 original reports (letters, case reports or full-length articles) published since 1 January 1970, which addressed 507 (350 males and 157 females) individually documented cases: 8 neonates less than 4 weeks of age, 251 infants less than 12 months of age, 182 infants 12 to 23 months of age and 65 children 24 months or more of age (this information was not available in one case). In all patients, the diagnosis of AHE made in the original reports was reviewed using three well-established clinical criteria: raised annular or nummular eruptions and inflammatory skin edema (mostly non-pitting, tender and sometimes also warmth) in a not-ill-appearing child [1,3,8,10]. The clinical diagnosis was supported by a skin biopsy disclosing a non-granulomatous neutrophil infiltration into small-vessel walls with karyorrhexis in 248 (50%) cases.

### 2.2. Eligibility Criteria

For the present analysis, we initially searched for the terms “Köbner”, “Köbnerization”, “non-blanching linear skin lesions” and “Pastia” in all fields of the publications. We also speculated the existence of images depicting, in addition to the typical rash, either a Köbner or a Pastia sign not addressed in the legend of the figures or in the body of the manuscript. Hence, we decided to sort all published cases of AHE which were supported by at least one photograph. Two of us meticulously assessed separately but in a non-blinded fashion the images for the existence of either a Köbner sign with an approximate length of ≥20 mm or Pastia linear skin fold lesions. Discrepancies were solved by consensus and, if needed, by consultation with a senior author.

The following information was extracted for each AHE case documented by at least one picture: demographic data, preceding (≤10 days) infection or vaccine, uncommon features (1. systemic features such as articular, abdominal, or kidney involvement; 2. eruptions such as blistering lesions or extensive skin necrosis; 3. production of tears tinged with blood (i.e., hemolacria); 4. compartment syndrome of the extremities; and 5. positive family history; that is, AHE or another vasculitis in a first-degree relative of a patient), and abnormally long (>3 weeks) disease duration.

### 2.3. Analysis

Continuous data are presented as median and interquartile range and were analyzed using the Kruskal–Wallis test. Categorical data are presented as frequency and were analyzed using the Fisher exact test. Significance was set at *p* < 0.05.

## 3. Results

Among the 507 cases, 339 were reported with and 168 without the support of at least one photograph, as shown in Figure 2. Cases with and without the support of a photograph did not significantly differ with respect to gender ratio (236 males and 103 females versus 114 males and 54 females) and age (11 (8–18) versus 12 (9–19) months of age; median and interquartile range).

The distinctive linear lesions of the Pastia or Köbner signs occurred in 83 cases (24%) that were reported between 1974 and 2021 in 79 articles [11,12,13,14,15,16,17,18,19,20,21,22,23,24,25,26,27,28,29,30,31,32,33,34,35,36,37,38,39,40,41,42,43,44,45,46,47,48,49,50,51,52,53,54,55,56,57,58,59,60,61,62,63,64,65,66,67,68,69,70,71,72,73,74,75,76,77,78,79,80,81,82,83,84,85,86,87,88,89]: 12 from Turkey, 8 from the United States of America, 6 from France, 6 from Spain, 5 from India, 5 from Italy, 5 from Portugal, 4 from Argentina, 3 from Brazil, 2 from Germany, 2 from Holland, 2 from Peru, and one each from Australia, Belgium, Chile, Columbia, Cyprus, Iran, Ireland, Japan, Korea, Lebanon, Malaysia, Maroc, Mexico, Paraguay, Qatar, Switzerland, South Africa, United Kingdom, and Venezuela. A total of 43 papers were published in English, 14 in Spanish, 7 in French, 5 in Portuguese, 5 in Turkish, 2 in Dutch, 2 in German and 1 in Italian. The terms Köbner or Pastia were used for 3 of the 83 cases [46,50,83].

A Pastia sign was noted in 51 [2,11,12,13,15,16,17,18,19,20,22,23,25,26,28,30,32,33,34,35,36,38,39,41,42,44,46,48,51,52,53,55,56,57,58,62,64,68,69,70,71,73,74,75,76,77,82,83,87,88], a Köbner sign in 24 [14,21,23,27,31,37,40,43,45,47,49,50,54,59,61,66,69,72,78,80,83,84,85,86] and both Pastia and Köbner signs in 8 [24,60,63,65,67,79,81,89] cases. Patients without and with Pastia, Köbner or both Pastia and Köbner signs did not significantly differ with respect to male to female ratio, age, prevalence of precursors, uncommon features, or long disease duration (Table 1).

The Köbner sign occurred in a total of 32 cases, comprised of 21 males and 11 females 11 [8,9,10,11,12,13,14,15,16,17] months of age. A total of 42 Köbner signs (Table 2) were documented in the mentioned 32 cases [14,21,23,24,27,31,37,40,43,45,47,49,50,54,59,60,61,63,65,66,67,69,72,78,79,80,81,83,84,85,86,89]. Lower legs, thighs, waistline, and groins were the most affected regions.

A total of 69 Pastia signs were documented in 59 patients (46 males and 13 females, 11 [7,8,9,10,11,12,13,14] months of age), as shown in Table 3 [6,8,11,12,13,15,16,17,18,19,20,22,23,24,25,26,28,29,30,32,33,34,35,36,38,39,41,42,44,46,48,51,52,53,55,56,57,58,62,63,64,65,67,68,69,70,71,73,74,75,76,77,79,81,82,87,88,89]. The ankles, feet, cubital fossa, and elbow were the most affected regions for this sign. Of note, 2 out of the 59 cases with Pastia signs were associated with a Mycoplasma pneumoniae upper respiratory infection [39,88] and 1 was associated with streptococcosis [35].

## 4. Discussion

This is the first systematic assessment of the prevalence of Köbner and Pastia signs (Figure 3) in AHE. The results may be recapitulated in three points. First, these signs occur in approximately every fourth case. Second, the disease course of this benign and self-limited childhood vasculitis is similar in cases without and with Köbner or Pastia sign. Third, the physical examination of children suspected to suffer from AHE focuses on the general appearance and the distinctive rash but disregards the Köbner and the Pastia signs, two of the most notorious skin signs. Hence, AHE should be added to the list of conditions that may present with these clinical skin signs.

The Köbner sign is a well-recognized clinical sign that is mainly caused by mechanical friction [4]. The localization we noted in this review and the patients’ age suggest that in AHE, this sign is mainly brought about by diapers and socks. Our findings are in line with the literature. The Köbner sign occurs in approximately 10% of patients with a skin-limited or skin-predominant vasculitis [4]. Furthermore, this sign has been recently identified in approximately every fourth child with Henoch–Schönlein purpura, the most common childhood vasculitis [90]. Controversy exists as to whether hemorrhagic edema of young children represents the infantile variant of Henoch–Schönlein purpura or whether it is a similar but distinct entity. The lack of systemic involvement, the absence of immunoglobulin A deposits in most cases, and the short, very benign course are strong arguments in favor of the latter hypothesis [1,2,3].

Probably, the most unexpected and surprising finding in this study relates to the fact that the Pastia sign is rather frequent in AHE. This sign characteristically occurs in streptococcal and staphylococcal scarlet fever and, less frequently, in eruptions caused by Mycoplasma pneumoniae [5,6]. There is no common association between Mycoplasma pneumoniae or Streptococcus A and AHE because this microorganism usually causes infections in children 4 years or more of age [5,91,92]. Remarkably, AHE was associated with Mycoplasma pneumoniae or Streptococcus A in three of the cases with a Pastia sign. We do not have any clear-cut explanation for the occurrence of the Pastia sign in both scarlet fever and AHE. However, an impaired capillary permeability secondary to the leukocytoclastic vasculitis plays a crucial role in the development of the skin features noted in AHE [1,2,3]. Similarly, capillary hyperpermeability is notoriously essential to produce the distinctive eruption of scarlet fever [5,93].

Most images included in reports addressing children with AHE did not include all skin regions and mainly concentrated more on annular or nummular eruptions than on Köbner or Pastia signs. Hence, we speculate that the present data likely underestimate the prevalence of these signs in hemorrhagic edema.

The results of the present review confirm that AHE is often preceded by a simple viral infection or by a vaccine [1,2,3,4]. Cutaneous vasculitides have been associated with the severe acute respiratory syndrome coronavirus 2 [94]. Unsurprisingly, therefore, one case included in this review was observed in a child affected by this infection [89].

In AHE, the skin lesions may present dramatically and are frequently initially misdiagnosed as an invasive bacterial infection (such as meningococcal septicemia), nonaccidental skin bruising or urticaria. The non-toxic appearance of affected children argues against the diagnosis of invasive bacterial infection, whereas Köbner or Pastia signs argue against the diagnosis of nonaccidental skin bruising or urticaria.

## 5. Conclusions

This is the first systematic assessment of the prevalence of Köbner and Pastia signs in AHE. This study points out that these classical skin signs occur in approximately every fourth case. Hence, the systematic assessment of these signs might help differentiate this vasculitis from other conditions with similar lesions. Finally, children with this vasculitis should not wear excessively tight clothes during the acute phase of the disease to limit the severity of the Köbner sign.

## Figures and Tables

**Figure 1 children-09-00265-f001:**
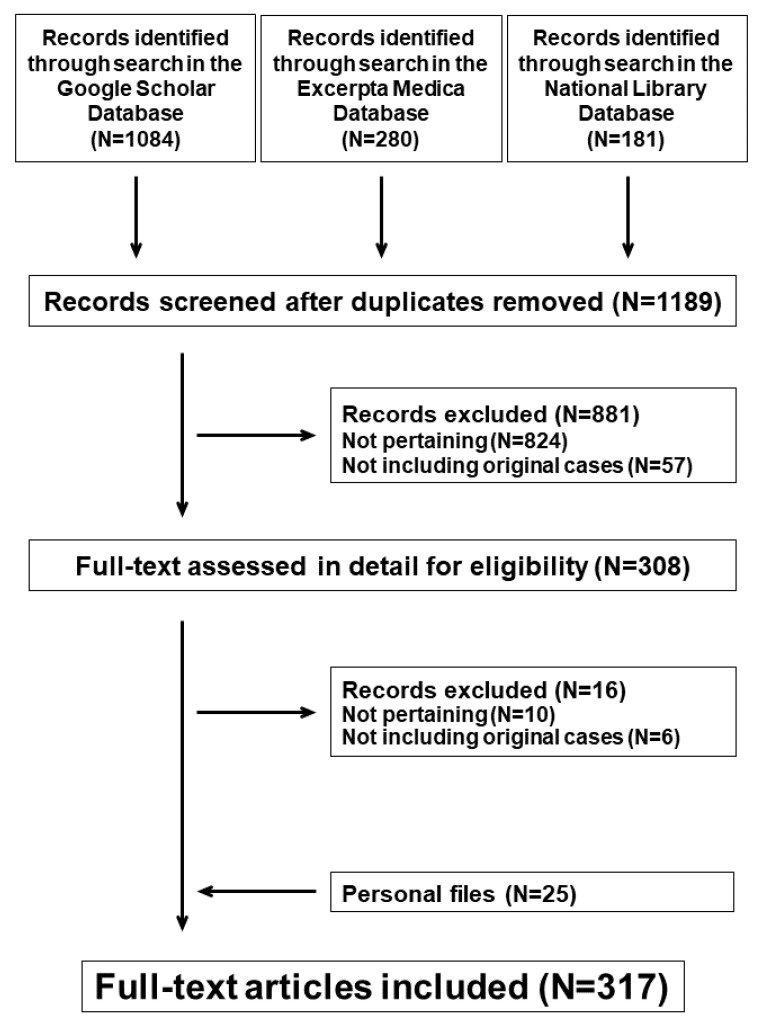
AHE. PRISMA flow diagram of the literature search process.

**Figure 2 children-09-00265-f002:**
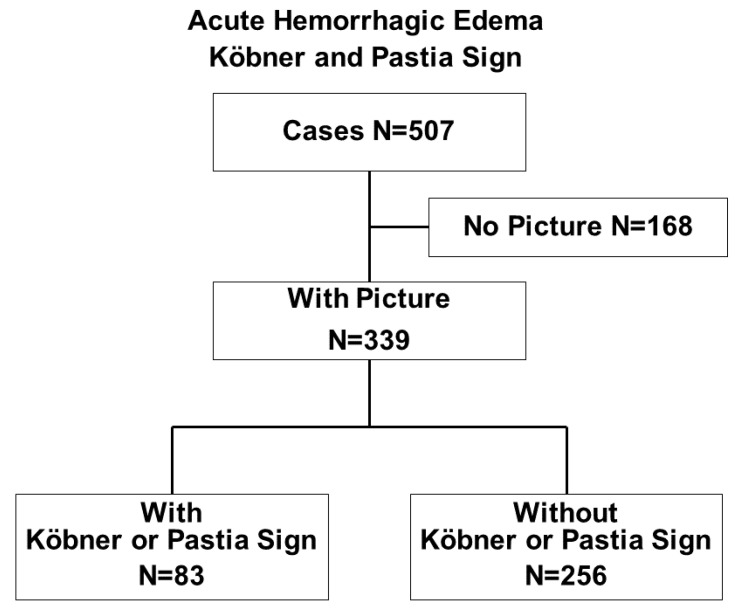
AHE presenting with Köbner or Pastia signs. This vasculitis syndrome was documented in 507 cases published between 1 January 1970, and 1 July 2021. The Köbner or the Pastia signs were noted in 83 of the 339 cases.

**Figure 3 children-09-00265-f003:**
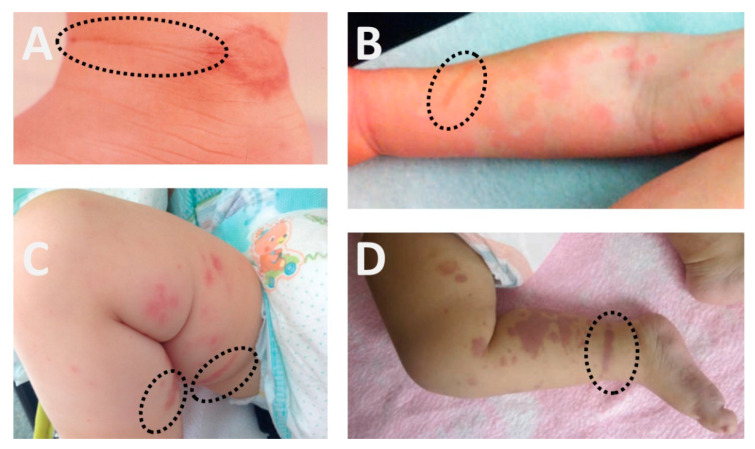
Pastia (panel **A**) and Köbner (panel **B**–**D**) signs in AHE.

**Table 1 children-09-00265-t001:** Characteristics of 339 AHE patients with and without Köbner or Pastia sign.

		without	with			
			Köbner or Pastia Sign	Köbner and Pastia Signs	Köbner Sign	Pastia Sign
N		256	83	8	24	51
Males:females, N		176:80	60:23	7:1	14:10	39:12
Age, months		11 [8,9,10,11,12,13,14,15,16,17,18]	11 [8,9,10,11,12,13,14,15,16]	11 [7,8,9,10,11,12,13,14,15,16,17]	11 [8,9,10,11,12,13,14,15,16,17]	11 [8,9,10,11,12,13,14,15]
Precursor						
	Infection, N	222	69	7	20	42
	Vaccine, N	13	2	1	0	1
Uncommon features, N		56	14	3	0	8
Long disease duration, N		4	0	0	0	0

**Table 2 children-09-00265-t002:** Localization of Köbner sign in 32 patients with AHE (21 males and 11 females, 11 [8,9,10,11,12,13,14,15,16,17] months of age). The Köbner sign was observed in two different regions in one case.

Localization	Unilateral	Bilateral	Total
Lower legs, N	10	6	16
Thigh, waistline, groin, N	9	1	10
Arms, N	4	0	4
Back, N	1	0	1
Face, N	1	0	1
Feet, N	1	0	1

**Table 3 children-09-00265-t003:** Localization of Pastia sign in 59 patients with AHE (46 males and 13 females, 11 [7,8,9,10,11,12,13,14] months of age). The sign was observed in 2 different regions in 10 cases [17,19,24,26,35,50,60,63,73,85].

Localization	Unilateral	Bilateral	Total
Ankle, feet, N	28	3	31
Cubital fossa, ellbow, N	9	1	10
Thigh, waistline, groin, N	6	2	8
Back, knee, popliteal fossa, N	6	1	7
Wrist, N	7	0	7
Axilla, N	4	2	6

## Data Availability

Data are available upon reasonable request to the corresponding author.

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
