# Peer review of "Köbner and Pastia Signs in Acute Hemorrhagic Edema of Young Children: Systematic Literature Review"

_children, 2022, doi:10.3390/children9020265_

Round 1

Reviewer 1 Report

Review

This is a systematic review to investigate Köbner and Pastia signs in young children with acute hemorrhagic edema (AHE).

Specific comments:

Acute hemorrhagic edema (AHE) is a benign skin-limited disease/vasculitis. Diagnosis could be made according to the clinical features/presentation with or without skin biopsy or laboratory examination. The authors should clarify why using Köbner and Pastia signs to define this disease. Clinical significance of Köbner and Pastia signs should be further emphasized in the manuscript.

1) In Acute Hemorrhagic Edema BIbliographic Database (AHEBID), search engines (Excerpta Medica, the National Library of Medicine databases and Google scholar) are screened every second month using the PRISMA guidelines. The authors should provide PRISMA guideline checklist to confirm every step the author did.

2) (Page 2 Line 66) The database included 317 original reports (letters, case reports or full-length articles) published since January 1, 1970. Was it possible for the authors miss some image galleries or snapshots, e.g. CMAJ October 26, 2020 192 (43) E1309; DOI: https://doi.org/10.1503/cmaj.200418., in the search results.

3) (Page 3 Line 86) Two of us meticulously assessed separately but in a non-blinded fashion the images for the existence of either a Kobner sign with an approximate length of ≥20 mm or Pastia linear skin fold lesions. How did the author resolve if discrepancies of assessment in individual result.

4) Diagnosis could be made according to the clinical features/presentation with or without skin biopsy or laboratory examination. The authors should clarify why using Köbner and Pastia signs to define this disease. Clinical significance of Köbner and Pastia signs should be further emphasized in the manuscript.

5) Similar articles about clinical presentations in AHE were published in J Am Acad Dermatol. (DOI: https://doi.org/10.1016/j.jaad.2020.12.033) and Dermatology (DOI: 10.1159/000519009).

Author Response

Re.: Köbner and Pastia signs in acute hemorrhagic edema of infancy: systematic literature review (Bronz et al.)

Dear Editor

Many thanks for your recent communication and for the opportunity to submit a revised version of our manuscript.

The manuscript was modified as follows.

1) Comments raised by the Editor: The length of your present version is a little shorter than what we expected for review paper. In general, systematic and comprehensive review papers will attract more readers and citations. So to make the paper more comprehensive and to attract more readers, we kindly suggest you to include more recent research results to enrich its content.

  • Many thanks for this interesting comment. In the revised version of the manuscript (section: Introduction) we provide data concerning the involvement of kidney, joints, and abdomen. We also add some information on the history of this vasculitis. The current manuscript is longer by slightly more than 100 words.

Comments raised by Reviewer #1

1) Acute hemorrhagic edema (AHE) is a benign skin-limited disease/vasculitis. Diagnosis could be made according to the clinical features/presentation with or without skin biopsy or laboratory examination. The authors should clarify why using Köbner and Pastia signs to define this disease. Clinical significance of Köbner and Pastia signs should be further emphasized in the manuscript.

  • Many thanks for this interesting and clinically relevant comment. The clinical relevance of Köbner of and Pastia signs is addressed in the section Discussion and Conclusions of the manuscript. In the revised version of the manuscript, we provide some more comments on this issue.

2) In Acute Hemorrhagic Edema BIbliographic Database (AHEBID), search engines (Excerpta Medica, the National Library of Medicine databases and Google scholar) are screened every second month using the PRISMA guidelines. The authors should provide PRISMA guideline checklist to confirm every step the author did.

  • This comment is constructive. In the revised version of the manuscript, we adjust this section of the manuscript (for example by citing the 2020 PRISMA guidelines). Furhermore, we provide a new reference, which details the PRISMA checklist and we add in the supplementary material our PRISMA checklist.

3) (Page 2 Line 66) The database included 317 original reports (letters, case reports or full-length articles) published since January 1, 1970. Was it possible for the authors miss some image galleries or snapshots, e.g. CMAJ October 26, 2020 192 (43) E1309; DOI: https://doi.org/10.1503/cmaj.200418., in the search results.

  • Many thanks for the comment. Our literature search was very careful. For example, we included 25 cases found exclusively in the references of articles detected by means of the literature search or in personal files. Furthermore, we also included some cases images presented exclusively as Quiz.

4) (Page 3 Line 86) Two of us meticulously assessed separately but in a non-blinded fashion the images for the existence of either a Köbner sign with an approximate length of ≥20 mm or Pastia linear skin fold lesions. How did the author resolve if discrepancies of assessment in individual results.

  • In the revised version of the manuscript we add this information.

5) Diagnosis could be made according to the clinical features/presentation with or without skin biopsy or laboratory examination. The authors should clarify why using Köbner and Pastia signs to define this disease. Clinical significance of Köbner and Pastia signs should be further emphasized in the manuscript.

  • The clinical impact of Köbner of and Pastia signs is more extensively addressed in the section Discussion and Conclusions of the revised manuscript.

6) Similar articles about clinical presentations in AHE were published in J Am Acad Dermatol. (DOI: https://doi.org/10.1016/j.jaad.2020.12.033) and Dermatology (DOI: 10.1159/0005190

  • Many thanks. The reports mentioned by the Reviewer are included in our reference list (reference #3 and #4).

Reviewer 2 Report

The article deals with acute hemorrhagic oedema that mainly affects children aged 2 years and younger thus investigating the prevalence of Köbner and Pastia signs. The topic is very interesting and represents an innovative point of view of the disease. The paper is complete, well designed and structured. Data analysis is accurate, detailed and results fully support authors’ conclusions. 

I have some minor criticisms:

1)    Use the abbreviation for acute hemorrhagic edema, avoiding redundance in the rest of the text

2)    With regard to Eligibility criteria the authors should use the terms Köbner, Köbnerization, non-blanching linear skin lesions and Pastia in quotes

3)    In table 1 the authors should clarify what “long disease duration” means (Months? Years?)

4)    Page 5 line 126: “63.” should be replaced with “63,”

5)    In table 2 and 3 the exact number of males and females should be provided

Kind regards

Author Response

Re.: Köbner and Pastia signs in acute hemorrhagic edema of infancy: systematic literature review (Bronz et al.)

Dear Editor

Many thanks for your recent communication and for the opportunity to submit a revised version of our manuscript.

The manuscript was modified as follows.

Comments raised by Reviewer #2

1) Use the abbreviation for acute hemorrhagic edema, avoiding redundance in the rest of the text

  • Many thanks for this remark. We used the abbreviation “AHE” for Acute Hemorrhagic Edema.

2) With regard to Eligibility criteria the authors should use the terms Köbner, Köbnerization, non-blanching linear skin lesions and Pastia in quotes.

  • We wrote the above-mentioned terms in quotes.

3) In table 1 the authors should clarify what “long disease duration” means (Months? Years?)

  • Many thanks for this comment. The meaning of “long disease duration” is reported in the section “Methods – Eligibity criteria”, line 100.

4) Page 5 line 126: “63.” should be replaced with “63,”

  • Many thanks for this correction, we corrected it.

5) In table 2 and 3 the exact number of males and females should be provided

  • Many thanks, the number of males and females is reported in the legend of the tables 2 and 3.

In conclusion, I hope that the revised manuscript is now considered acceptable for publication in “Children”. Please do not hesitate to contact me again if you feel that further corrections are required.

Sincerely Yours,

Gabriel Bronz

Round 2

Reviewer 1 Report

Thank you for the thorough explanations. The authors have correctively responded to Editor and Reviewers comments.